# Medicine manipulation: An alternative to mitigate therapeutic gaps in the Brazilian Unified Health System?

Gabriel Gonçalves Okamoto[1], Kathiely Martins dos Santos[2], Luma de Lira Nogueira[3], Guilherme Martins Gelfuso[3], Rafael Santos Santana[3]*

1 Department of Pharmaceutical Services and Strategic Inputs, Brazilian Ministry of Health, Brasília, Federal District, Brazil, 2 Department of Public Health, University of Brasília, Brasília, Federal District, Brazil, 3 Department of Pharmacy, University of Brasília, Brasília, Federal District, Brazil

* rafael.santana@unb.br

**Data Availability Statement:** All relevant data are within the paper.

**Funding:** The author(s) received no specific funding for this work.

## Abstract

Despite the various initiatives carried out in Brazil and in the world, the challenge of offering essential medicines in adequate presentations remains, especially to the public affected by diseases considered neglected and the pediatric population, for whom the therapeutic options remain limited. The main objective of this study was to evaluate the production of manipulated medicines as a strategy to mitigate therapeutic and access gaps to essential medicines within the Brazilian public health system, called the Unified Health System (SUS). The evaluation, carried out between 2020 and 2021, identified, among the medicines considered essential to the Brazilian health context, those unavailable, for which strategies were evaluated to mitigate the identified unavailability, which is conventionally called therapeutic gaps. For 57% (n = 235) of pharmaceutical presentations identified as therapeutic gaps in SUS, manipulation was identified as the best strategy to promote access. Of these presentations, 30% (n = 70) were identified as priorities in the context of patient care and were mainly related to the demands of the pediatric public and those affected by poverty-related diseases. Concerning poverty-related diseases, the absence of evidence on the development of a standard formula for drugs with indication for such diseases was demonstrated. The need for an annual investment of approximately US$74.75 per capita was estimated to offer treatments in adequate presentations to SUS users, which should reflect in the improvement of the quality of life of about 26 thousand people. It was observed that this investment amount corresponds to only 3% of the budget for the purchase of medicines financed exclusively by the Ministry of Health thorugh the Strategic Component of Pharmaceutical Assistance (CESAF) approved for 2021.

## Introduction

The limitations of available treatments for diseases considered neglected or that affect the most vulnerable populations, such as tuberculosis, leishmaniasis and sickle cell disease, have been discussed for decades [1, 2]. It is estimated that one billion people around the world are

**Competing interests:** The authors have declared that no competing interests exist.

susceptible to illness from these diseases [3]. The limitation of adequate drug presentations also affects pediatric patients, tube users, and those with swallowing difficulties. This group of patients are considered 'therapeutic orphans' in terms of the availability of presentations capable of meeting their specific needs, as Captopril, Isoniazid, Primetamine and Sulfadiazine drugs, generally prescribed in pediatrics, but available only in the form of oral solids in Brazil [4].

In fact, the absence of adequate treatments affects the pediatric population worldwide [5–8]. Two-thirds of this population use drugs that do not have specific indications for this age group [8].

In Brazil, the National Health Surveillance Agency (ANVISA) is the responsible for authorizing drug registrations in the country. However, to date, there is no specific regulation for drugs intended for the pediatric population, nor are there any evidence-based guidelines on the transformation of pharmaceutical forms for the administration of adequate doses [9, 10] This reality makes the use of medication less safe, making it difficult to predict and reliably treat therapeutic results [9, 10].

Despite its relevance to public health, neither the pharmaceutical industry nor public and private research centers have invested sufficiently in improving the technologies used to care for this population or in the development of new drugs [3]. Only 7% of research and development (R&D) projects from twenty global pharmaceutical companies target pediatric patients, which reveals a large gap in the pediatric drug pipeline [11].

Although in Brazil some mechanisms seek to ensure universal access to essential medicines, as is the case with both the establishment of the National List of Essential Medicines (Rename) and the adhesion to the Strategic Fund of the Pan American Health Organization (PAHO), a mechanism which guarantees the exceptional importation of medicines without registration by the National Health Surveillance Agency (Anvisa), in addition to the publication of Ordinance GM/MS No. 2,531/2014, which determines the criteria for defining the list of strategic products for the public health system system, called the Single Health System (SUS) [12, 13], therapeutic gaps persist and reinforce the need for the country to move towards its technological independence, aiming to solve its social demands [12].

Although access to compounded medicines is not covered by SUS, this study evaluated the adoption of this strategy to mitigate existing therapeutic gaps. The study conceptualized a compounded medicine as one obtained through the manipulation of any masterful formula or officinal preparation, prescribed, obtained and dispensed according to individual needs, which can be produced in pharmacies, universities, hospital pharmacies, among other structures, as long as they follow good manufacturing practices. Hereafter, we shall refer to this process as magistral manipulation, that is, magistral drug production.

The technical-operational and economic feasibility was mainly evaluated [9], carrying out, for this purpose, several surveys that ranged from the identification of essential drugs in the Brazilian sanitary context to the evaluation of the adequacy of pharmaceutical presentations covered by SUS. From this, incorporations of appropriate presentations were proposed and the technology necessary for its development, the feasibility of masterful production, and the investment required were identified.

## Methods

The methodological path of this work comprised analyzes, in stages, that sought to identify essential drugs in need of adjustments for more adequate pharmaceutical presentations considering therapeutic convenience, dosage and target audience, as established by the Ministry of Health guidelines. We sought to investigate the commercial availability of these drugs in Brazil

and in SUS, in addition to presenting access strategies to mitigate the identified therapeutic gaps.

## Identification of essential medicines in appropriate presentations

**Identification of the list of essential drug products.** The first stage consisted of identifying drugs common to the National List of Essential Medicines (Rename) [14] and the Word Health Organization Model List of Essential Medicines, 21st edition, 2019 ("WHO List") [15]. Once identified, it was verified whether there was any mention of these medicines in guidelines and guides prepared by the Ministry of Health that addressed therapeutic recommendations, in order to identify essential medicines in the Brazilian health context [16–18].

**Identification of suitable presentations.** Subsequently, based on the aspects of indication for use, dosage and target audience, as indicated in the guidelines and guides consulted, the drugs were analyzed in regards to adequacy of the pharmaceutical form, with adequate presentations being considered those that met the criteria of "dosage convenience" and "appropriate pharmaceutical presentation". In this context, "dosage convenience" was understood as the guarantee of the use of pharmaceutical presentations that allow, at least, the fulfillment of the therapeutic protocol with as few administrations as possible and without the need for fractionation or adaptations of the pharmaceutical form.

**Research on commercial availability and SUS coverage.** From this survey, an investigation was carried out on the commercial availability in Brazil and SUS coverage of essential medicines in adequate presentations, based on information regarding health registration occurrence and/or market regulation, which was obtained through consultations on Anvisa's website [19, 20].

**Identification of therapeutic gaps.** Subsequently, the "therapeutic gaps existing in SUS" were considered as: (i) the presentations commercially available in the country and not covered by SUS; (ii) presentations commercially unavailable in the country and covered SUS, or (iii) those presentations commercially unavailable in the country and not covered by SUS. All presentations included in Rename were considered covered by SUS.

**Identification of medicines that can be magistrally produced by manipulation.** Pharmaceutical presentations commercially unavailable in the country were categorized according to the complexity of the technologies and processes necessary for their development into high and low complexity. Presentations whose development involves costly processes and productions and which demand high investments in the long term were considered highly complex, such as injectables and other sterile presentations. Presentations not meeting these criteria were considered of low complexity and identified as presentations that could be magistrally produced via manipulation.

## Prioritization of SUS demands in regards to medicines that can be nationally produced through manipulation

Through Law No. 12,527/2011, known as the Access to Information Law, which regulates the constitutional right of access to public information [21], a diagnosis was obtained on the prioritization of the supply of essential medicines unavailable in SUS and that can be manipulated.

For this analysis, presentations that were in demand and that were used in diseases and conditions in which the absence of available therapeutic alternatives was verified were considered as priorities, taking into account the adequacy of the presentation to the target audience. Thus, if at least 1 (one) state indicated a demand for the drug, it would then be considered a "priority in the context of patient care", and the item would then be included in the final priority list.

## Assessment of the feasibility of national production of essential medicines in adequate presentations via magistral manipulation within the scope of SUS

**Assessment of theoretical feasibility.** Once the priority list was established, the theoretical feasibility was evaluated, in which the existence, in the literature, of the standard formula for the presentations prioritized in the previous stage was considered. For this purpose, searches were carried out in the Brazilian Pharmacopeias, 6th edition [22] and in the US Pharmacopeia National Formulary, USP 34 NF 29 [23]. In case of lack of information in the main sources, additional searches were carried out in books and articles related to the theme [24–29].

The search results were categorized into: (i) standard formula described in the literature and (ii) standard formula not described in the literature. The presentations not described in the consulted literature were considered without theoretical feasibility, being excluded from the next stages of the analysis.

**Assessment of magistral poduction's feasibility.** The feasibility of magistral productionassessment was carried out by identifying the availability, in the Brazilian market, of active principles that make up the priority formulations and which had theoretical feasibility.

During the months of February to April 2021, the main national suppliers were consulted about information on the value in reais and the minimum commercialized quantity of the main actives involved in the formulations. Concomitantly, the values of packaging, capsules, among other relevant inputs for the survey of production costs were consulted.

Considering the specifics regarding the measurement units adopted by the suppliers, all values were, when necessary, converted to price in reais per gram of excipient. The information obtained was systematized and the presentations identified as: (i) active ingredient commercially available in the country; or (ii) active ingredient commercially unavailable in the country.

In case of lack of information on the national commercialization of the active ingredient, complementary searches were carried out in order to identify the commercialization of the finished product in the national market.

In cases of unavailability of the active ingredient or finished product in the national market, the presentation was considered unfeasible from the point of view of magistral production, being excluded from the analysis.

## Demand and cost estimates for the incorporation of prioritized medicines in SUS

**Estimeted cost.** In order to estimate costs for the magistral production of prioritized drugs, which presented theoretical and production feasibility, data on the average price of active ingredients were used as well as information obtained from the literature on production cost.

The values of the active ingredients were obtained through quotations requested from suppliers in the national market. In case of unavailability of suppliers, a survey was carried out in the Price Database or in the Price Panel [30, 31].

In case oflack of information on the values, or of the commercialization of the active ingredient in Brazil, the value of the finished product was considered, also obtained through a survey in the Price Database, as well as in the Price Panel of the Ministry of Health, using the other previously informed costs [30, 31].

Pricing criteria were adopted in accordance with the parameters of the National Association of Magistral Pharmacists (Anfarmag) and a study by Michels et al. (2017), which

comprises direct costs, indirect costs and operating costs. Sales costs were not considered in these calculations, as there is no interest in obtaining a profit [32, 33].

Direct costs included the cost of the active ingredient, the cost of primary packaging, and the price of labor, which represented the cost of handling a base formulation. The indirect and operational costs were considered as being 10% and 19% of the final value of the formulation, respectively [32, 33].

Indirect costs refer to the amount invested in the maintenance of the magistral drug production laboratory, for example, rent, energy, personal protection equipment, quality control, waste collection, prevention of environmental risks, water, maintenance and cleaning material. Operating costs, on the other hand, are those that involve a monthly fee for the system, telephone, internet, office supplies and consumption and accounting fees [32, 33].

For this study, neither the cost of taxes nor the cost of sales were added, which generally takes into account different factors, such as the consuming public, competition and the market, generally marked by the multiplication of the cost price of the raw material and packaging for a profitability value added to a fixed value.

**Estimated demand.**   To estimate demand, data on prevalence, incidence or, when unavailable, number of cases in 2019, of diseases and conditions for which prioritized presentations have the main indication were considered, according to the age group of the target audience. Data on the number of cases, prevalence or incidence were obtained by the Access to Information Law and complementary searches were carried out in the official domains of the Ministry of Health.

**Estimated cost for incorporation.**   Thus, the production calculation considered the average number of pharmaceutical units required for the monthly treatment of an individual, taking into account the dosages indicated in the guidelines and guides consulted. Additionally, the volume of 100 mL was standardized for oral liquids; for capsules, 60 capsules; and for creams and ointments, 100 g. By convention, even if the recommended treatment regimen provided for a quantity of less than 100 mL, 60 capsules or 100 g, these quantities were still considered to be the minimum necessary for the complete treatment of the patient. Regarding the average treatment time, the minimum time considered in this study was one month, even in the cases of drugs that have a treatment time shorter than this. In the case of medications of continuous use, the study considered the period of one year.

In cases of lack of information, both on values and on estimated public, the corresponding presentation was excluded from the analysis.

## Results

From the initial analyzes carried out in the present study, it was identified that 57% n = 235) of pharmaceutical presentations could be produced via magistral manipulation. These are drugs such as Artesunate oral solution, used to treat malaria or Hydrochlorothiazide 12.5 mg tablet, which despite being in the list of Rename, do not present valid health registration in the country.

Table 1 shows that of the total number of drugs that could be produced by magistral manipulation, when considering the classification (Anatomic Therapeuric Chemical–ATC), approximately 31% are related to the class of general antiinfectives for systemic use, antiparasitics products, insecticides and repellants, whose therapeutic indication is often associated with neglected or poverty-related diseases.

Other classes that showed great relevance according to the ATC classification were medications related to diseases of the alimentary tract and metabolism and the cardiovascular system. These are medications such as Pyridoxine (vitamin B6), Cholecalciferol, Spironolactone and

**Table 1. Summary of types of medicines with manufacturing potential in Brazil, through manipulation, according to the ATC classification (Brazil, 2021).**

| ATC GROUP | DESCRIPTION | QUANTITY OF MEDICINE | % OF MEDICINE ACCORDING TO ATC CLASSIFICATION |
|---|---|---|---|
| J | Antiinfective for systemic use | 50 | 21.30 |
| A | Alimentary tract and metabolism | 34 | 14.46 |
| C | Cardiovascular system | 25 | 10.63 |
| P | Antiparasitic products, insecticides, and repellents | 24 | 10.20 |
| N | Nervous system | 21 | 8.93 |
| L | Antineoplastic and immunomodulating agents | 16 | 6.80 |
| D | Dermatologicals | 13 | 5.60 |
| H | Systemic hormonal preparations (excluding sex hormones) | 9 | 3.82 |
| G | Genito urinary system and sex hormones | 9 | 3.82 |
| NC | Not included | 9 | 3.82 |
| B | Blood and blood forming organs | 8 | 3.40 |
| M | Musculo-skeletal system | 5 | 2.12 |
| HF | Herbal medicinal products | 5 | 2.12 |
| V | Various | 4 | 1.70 |
| R | Respiratory system | 3 | 1.28 |
| **TOTAL** | | **235** | **100.0** |

**Source:** Elaborated by the authors (2021).

*ATC = Anatomic Therapeutic Chemical

Captopril, in oral solution presentations, whose unavailability causes treatment difficulties, especially in the pediatric population, including newborns (Table 1).

From the drug products identified as plausible candidates for local manufacturing by manipulation, 30% were appointed as priority. They are medications such as Pyrimethamine, Captopril and Oseltamivir in oral solution presentations, which are used for treating Toxoplasmosis, Influenza and Systemic Arterial Hypertension, respectively.

The unavailability of pharmaceutical presentations suitable for the pediatric public was evidenced, in this study, by the finding that 88% of the drugs evaluated as priority for patient care were related to pediatric use indications, with 72% having an exclusive indication for this public. Only 10 of the prioritized presentations did not present previous studies for magistral manipulation in the presentations listed. It is noteworthy that of those presentations that do not have a standard formula described, nine are indicated for diseases related to poverty, such as Linezolid oral solution, used in the treatment of resistant tuberculosis, and Praziquantel oral solution, indicated in cases of schistosomiasis.

The evaluation of the profile of the priority drugs, according to the ATC classification, showed that 38% of the priority drugs belonged to the class of general anti-infectives for systemic use, antiparasitic, insecticides and repellents, whose therapeutic indication in 89% of the cases was related to poverty-related diseases.This study showed that of the 70 prioritized presentations, 73% have the commercialization of the active principle in the Brazilian market and 17% of the presentations, despite not having the commercialization of the active principle, can be obtained through the adaptation of the pharmaceutical form, as they present the drug in presentations different from those considered appropriate in the present study, such as Clomipramine used in the form of pills to treat obsessive-compulsive disorder, panic disorder, among others.

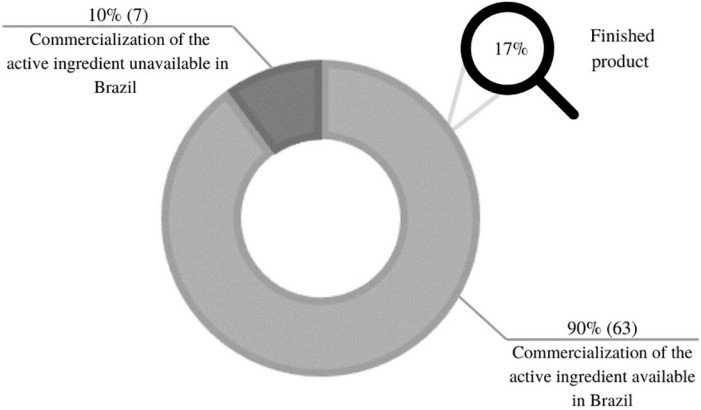

**Fig 1. Availability of active ingredients for the development of essential and priority manipulated medicines for patient care in SUS, Brazil, in 2021. Source:** Elaborated by the authors, 2021.

Another very important drug that was identified in this study as being subject to manipulation through the finished product is Hydroxyurea, used in the treatment of Sickle Cell Disease, which affects more than 25,000 people annually, mainly the black population and people in situations of social vulnerability [1].

As shown in Fig 1, only 10% of the prioritized presentations did not have the commercialization of the active ingredient, nor the finished product in different presentations in the national market. Thus, in addition to being considered unfeasible for magistral production, they were excluded from the cost estimate analysis, as the handling cost survey became unfeasible. Examples of these presentations are Pyrantel in oral solution and Ergocalciferol in capsule and oral solution presentations, respectively (Fig 1).

The present study also showed that 80% of the seventy prioritized drugs have production feasibility via manipulation because they have both the standard formula pre-defined in the literature, as well as the availability of the active ingredient, or of a finished product containing this active ingredient in the Brazilian market, as shown in Fig 2.

In Table 2, the drugs considered viable for development via magistral drug production are presented, as they have both the commercialization of the active ingredient in the national

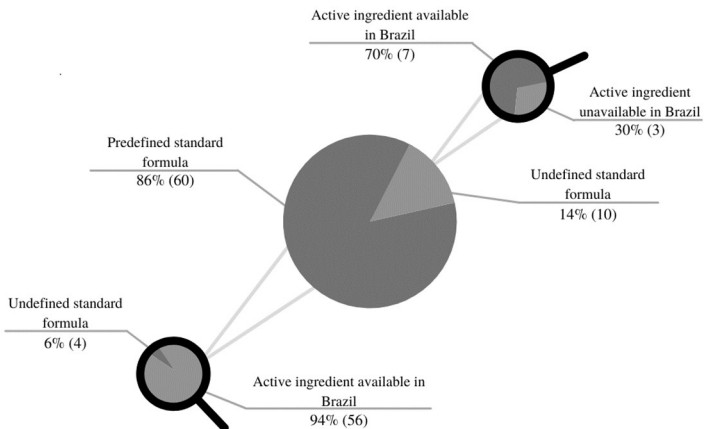

**Fig 2. Development feasibility via manipulation of priority drugs for patient care in SUS in Brazil in 2021. Source:** Elaborated by the authors, 2021.

**Table 2. List of viable drugs for development via manipulation in SUS (Brazil, 2021).**

| Drug | Presentation | Main indication of use | Use |
|---|---|---|---|
| Amlodipine | 2,5 mg/mL Oral solution | Hypertension. Stable and variant angina (prophylaxis). | Infantile |
| Benznidazole | 100 mg/5 mL Oral solution | American trypanosomiasis (Chagas disease). | Infantile |
| Betamethasone | 0.1% Lotion or ointment | Severe inflammatory skin conditions including contact dermatitis, atopy dermatitis (eczema), seborrheic dermatitis, lichen planus, and intractable pruritus; psoriasis of the scalp, hands and feet. | Both |
| Captopril | 1mg/mL Oral suspension | Systemic arterial hypertension, including hypertensive urgencies. Diabetic nephropathy; Left ventricular dysfunction after acute myocardial infarction; Congestive heart failure. | Infantile |
|  | 5 mg/mL Oral suspension |  |  |
| Calcium carbonate | 500 mg/5 mL Oral solution | Osteoporosis. Hyperphosphatemia associated with chronic kidney disease; Calcium deficiency. | Infantile |
| Calcium carbonate + cholecalciferol | 1.250 mg (500 mg of calcium) + 400 UI/5 mL Oral solution | Treatment and prevention of osteoporosis; Treatment of Paget's disease; Deforming osteitis. Treatment of osteogenesis imperfecta using pamidronate; Treatment of patients with multiple myeloma using zoledronic acid; Treatment of rickets. | Infantile |
| Activated charcoal | Powder for oral suspension | Nonspecific antidote used in acute exogenous intoxications by: acetylcysteine, phenothiazines, aspirin, mefenamic acid, indomethacin, valproic acid, iodides, amphetamines, ipecac, antidepressant agents, antimony, mercuric chloride, arsenic, morphine, atropine, methylene blue, opium, barbiturates, organophosphates, camphor, acetaminophen, carbamazepine, paraquat, chlordiazepoxide, parathion, chlorpheniramine, penicillin, chloroquine, silver, cocaine, primaquine, colchicine, probenecid, propoxyphene, propantheline, diazepam, mepacrine, digitalis, quinidine, strychnine, quinine, ethchlorvynol, methyl salicylate, phenylbutazone, selenium, sulfonamides, phenytoin, theophylline, phenol and tetracyclines. | Both |
| Clomipramine | 25 mg/mL Oral solution | Obsessive-compulsive disorder; Panic disorder; Major depressive disorder; Chronic pain; Paranoid schizophrenia. | Infantile |
| Chloramphenicol | 25 mg/mL Oral solution | Alternative treatment of serious infections, by susceptible bacteria; Typhoid fever; Meningeal plague; Brazilian spotted fever and other rickettsiosis; Secondary infection by bothropic accident. | Infantile |
| Clindamycin hydrochloride | 75 mg/5 mL Oral solution | Treatment of infections with susceptible organisms where there is penicillin allergy and resistance to first-line drugs, including staphylococcal bone and joint infections, peritonitis, and pneumonia. | Infantile |
| Ethambutol + Isoniazid + Rifampin | 275 mg + 75 mg + 150 mg Capsule | Tuberculosis, in combination with other drugs. | Adult |
| Dapsone | 25 mg/mL Oral solution | Leprosy treatment; Primary and secondary prophylaxis of Pneumocystis jirovecii pneumonia in co-infection with HIV (alternative regimen); Primary and secondary prophylaxis of Toxoplasma gondii, in co-infection with HIV (alternative regimen); Treatment of dermatitis herpetiformis. | Infantile |
| Chloroquine phosphate | 50 mg/5 mL Oral solution | Treatment of acute malaria caused by P. malaria, P. vivax and P. ovale (for infections caused by P. vivax and P. ovale, there is subsquent use of primaquine to eliminate intrahepatic forms); malaria prophylaxis; rheumatic disorders. | Infantile |
| Doxycycline | 25 mg/5 mL Oral solution | Bacterial infections; quinine or artesunate treatment supplement for multidrug treatment; P. falciparum malaria; short-term prophylaxis of multidrug-resistant P. falciparum Malaria. | Infantile |
| Enalapril | 5 mg/mL Oral solution | Hypertension; Cardiac insufficiency. | Infantile |
| Spironolactone | 10 mg/5 mL Oral solution | Systemic arterial hypertension; Congestive heart failure; Hirsutism. | Infantile |
| Ethambutol | 25 mg/mL Oral solution | Treatment of sensitive tuberculosis, in association with other antimicrobials. Treatment of drug-resistant tuberculosis in association with other antimicrobials. Treatment of tuberculosis in cases of intolerance, allergy or toxicity to the basic regimen, in association with other antimicrobials; Treatment of tuberculosis-HIV co-infection replacing the rifampicin regimen, when associated with a protease inhibitor or integrase inhibitor; Treatment and secondary prevention of Mycobacterium avium complex infection in HIV positive patients. | Infantile |
| Phenytoin | 20 mg/mL Oral suspension | Epilepsy–generalized tonic-clonic seizures, complex focal seizures, or a combination of both; Prevention and treatment of epileptic seizures during or after neurosurgical procedure; Treatment of tonic seizures, typical of Lennox-Gastaut syndrome; Status epilepticus. | Both |

(*Continued*)

**Table 2.** (Continued)

| Drug | Presentation | Main indication of use | Use |
|---|---|---|---|
| Fluconazole | 10 mg/mL Oral solution | Disseminated candidiasis; Candidiasis of the urinary tract; Esophageal candidiasis; Oropharyngeal candidiasis (treatment in patients with HIV); Vulvovaginal candidiasis; Coccidiomycosis (prophylaxis and treatment in HIV patients); Pulmonary cryptococcosis (treatment in HIV patients); cryptococcal meningitis; Prevention of fungal infections in patients undergoing bone marrow transplant. | Both |
| Fluoxetine | 20 mg/5 mL Oral solution | Depressive disorder; Obsessive-compulsive disorder; Panic disorder; Bipolar affective disorder type 1. | Infantile |
| Leucovorin | 5 mg/mL Oral solution | Prevention of toxicity or treatment of pyrimethamine overdose; Methotrexate high-dose rescue or overdose treatment. | Infantile |
| Fludrocortisone | 100 mcg/mL Oral solution | short-term suppression of inflammation in allergic diseases. | Infantile |
| Furosemide | 4 mg/mL Oral solution | Arterial hypertension; Refractory edema of different causes. | Infantile |
| Griseofulvin | 125 mg/5 mL Oral solution | Fungal infections of the skin, scalp or hair where topical treatment has failed or is inappropriate. | Infantile |
| Hydralazine | 5 mg/mL Oral solution | Severe and refractory systemic arterial hypertension; Hypertensive emergency. | Infantile |
| Hydrochlorothiazide | 5 mg/mL Oral solution | Systemic arterial hypertension; Edema of different causes. | Infantile |
| Hydrocortisone | 5 mg/mL Oral solution | Congenital adrenal hyperplasia; Addison's disease; adrenal hypoplasia; chronic maintenance or replacement therapy. | Infantile |
| Hydroxyurea | 100 mg/mL Oral solution | Sickle Cell disease; Chronic myeloid leukemia. | Infantile |
| Isoniazid | 50 mg/5 mL Oral solution | Treatment of sensitive tuberculosis, in association with other antimicrobials; Treatment of tuberculosis-HIV co-infection replacing the rifampicin regimen, when associated with a protease inhibitor or integrase inhibitor; Treatment of drug-resistant tuberculosis, in association with other antimicrobials; Treatment of drug-resistant tuberculosis, in association with other antimicrobials, in special situations (high doses);Treatment of tuberculosis in cases of intolerance, allergy or toxicity to the basic regimen, in association with other antimicrobials; Treatment of latent tuberculosis infection. | Infantile |
| Isoniazid + Rifampin | 75 mg + 150 mg Capsule | Treatment of sensitive tuberculosis; Tuberculosis in cases of intolerance, allergy or toxicity to the basic regimen, in association with other antimicrobials. | Adult |
| Levofloxacin | 250 mg/5 mL Oral solution | Treatment of drug-resistant tuberculosis in association with other antimicrobials. Treatment of tuberculosis in cases of intolerance, allergy or toxicity to the basic regimen, in association with other antimicrobials. | Infantile |
| Levothyroxine sodium | 25 mcg/mL Oral solution | Maintenance treatment in hypothyroidism; Suppression of thyroid-stimulating hormone (TSH) secretion in specific situations, such as differentiated thyroid carcinomas. | Infantile |
| Losartan potassium | 12.5 mg Capsule | Systemic arterial hypertension; Stroke prophylaxis in hypertensive patients with left ventricular hypertrophy; Diabetic nephropathy in patients with type 2 diabetes mellitus and a history of hypertension. Congestive heart failure. | Adult |
| | 12.5 mg/mL Oral solution | | Infantile |
| Mesalamine | 30 mg/mL Rectal enema | Crohn's disease; ulcerative colitis. | Adult |
| Metformin | 200mL/5mL Oral solution | Type 2 diabetes mellitus. | Infantile |
| Naproxen | 250 mg/mL Oral solution | Psoriatic arthritis; Rheumatoid arthritis; Ankylosing spondylitis. | Infantile |
| Nitrofurantoin | 5 mg/mL Oral solution | Treatment of urinary infections; Prophylaxis of recurrent urinary infections; Prophylaxis in genitourinary surgery. | Infantile |
| Omeprazole | 20 mg/sachet Powder for oral solution | Gastroesophageal reflux disease; acid-related dyspepsia; benign gastric ulcer treatment including those that complicate therapy with non-steroidal anti-inflammatory drugs (NSAIDs); Zollinger-Ellison syndrome; acid aspiration prophylaxis; fat malabsorption despite pancreatic enzyme replacement therapy in cystic fibrosis; Eradication of Helicobacter pylori. | Infantile |
| Ondansetron | 4 mg/5 mL Oral solution | Prophylaxis of nausea and vomiting induced by antineoplastic drugs with moderate and high emetogenic potential; Prophylaxis of post-surgical nausea and vomiting. | Infantile |
| Oseltamivir | 10 mg/mL Oral solution | Prevention and treatment of influenza virus infection. | Infantile |
| Vitamin A palmitate | 150,000 UI/mL Oral solution | Treatment of retinol deficiency; Prevention of complications (pneumonia and mortality) from measles in children under 2 years of age. | Infantile |
| Penicillamine | 250 mg/ 5 mL Oral solution | Wilson's disease; Cystinuria; Antidote for heavy metal poisoning (chelating agent for mercury, nickel, lead, arsenic and copper). | Infantile |

(*Continued*)

**Table 2.** (Continued)

| Drug | Presentation | Main indication of use | Use |
|---|---|---|---|
| Pyridoxine (vitamin B6) | 1 mg/mL Oral solution<br>10 mg/mL Oral solution | Prevention of drug-induced peripheral neuropathy (such as isoniazid, linezolid and terizidone); Wilson's disease; Prophylaxis and treatment of pyridoxine deficiency; Sideroblastic anemia. | Both |
| Pyrimethamine | 1 mg/mL Oral solution | Treatment of toxoplasmosis, in combination with sulfadiazine or clindamycin and leucovorin. | Infantile |
| Propylthiouracil | 50 mg/mL Oral solution | Hyperthyroidism; Treatment of hyperthyroidism; thyrotoxic crisis; thyrotoxicosis. | Infantile |
| Propranolol | 10 ml/mL Oral solution | Systemic arterial hypertension; Cardiac arrhythmias; Pheochromocytoma (only when associated with an alpha-blocker); Chronic angina pectoris; Prophylaxis after myocardial infarction; Essential tremor; Migraine prophylaxis; Infantile hemangioma; Idiopathic hypertrophic subaortic stenosis; Tetralogy of Fallot in children; Thyroid syndrome in children. | Infantile |
| Sildenafil citrate | 10 mg/mL Oral solution | Pulmonary arterial hypertension (PAH); Systemic sclerosis. | Infantile |
| Simvastatin | 10 mg/mL Oral solution | Dyslipidemias (hypercholesterolemia, hyperlipoproteinemia, hypertriglyceridemia, mixed hyperlipidemia). Prevention of cardiovascular events (patients at high risk of cardiovascular events and cerebrovascular accidents). | Infantile |
| Metoprolol succinate | 25 mg/mL Oral solution | Hypertension; Cardiac insufficiency; Stable chronic angina; Acute myocardial infarction. | Infantile |
| Sulfadiazine | 100 mg/5 mL Oral solution | Toxoplasmosis; Prevention of recurrence of rheumatic fever; Uncomplicated acute urinary tract infections. | Infantile |
| Hydroxychloroquine sulfate | 200 mg/5 mL Oral solution | Rheumatoid arthritis; Dermatomyositis and polymyositis; Systemic lupus erythematosus. | Infantile |
| Topiramate | 25 mg/mL Oral solution | Epilepsy–monotherapy in focal or primary generalized tonic-clonic seizures (monotherapy); Epilepsy–adjuvant therapy in focal seizures, primary generalized seizures or those associated with Lennox-Gastaut syndrome. | Infantile |
| Warfarin sodium | 0.5 mg/mL Oral solution | Treatment and prevention of thromboembolic disorder in atrial fibrillation and rheumatic heart disease; Prevention of thrombosis after myocardial infarction and after prosthetic heart valve insertion. Prevention of recurrence of myocardial infarction and venous thromboembolism; Treatment and prevention of pulmonary embolism and transient ischemic attack. | Infantile |

**Source:** Elaborated by the authors, 2021.

market and the standard formula available in the literature, totaling 56 drugs used, in the vast majority, by children.

For notifiable diseases, searches were carried out in the Notifiable Diseases Information System (SINAN), via Tabnet, and data referring to 35% (n = 25) of the prioritized presentations were located. For the other presentations, reliable data were not obtained for the demand estimate survey, and, therefore, the presentations were excluded from the cost estimate analyses.

Table 3 shows the annual cost estimate stratified by prioritized drug, in relation to the investment for its development via manipulation, through local manufacture in pharmacies, universities, hospital pharmacies, among other structures appropriate for good manufacturing practices.

The production cost estimate of 100% (n = 56) of the presentations considered viable, that is, whose presentations have both the standard formula pre-defined in the literature, and the availability of the active ingredient or finished product in the Brazilian market, totaled the average production value of approximately $7.65 per bottle.

Taking into account only the production prices, demand estimates and average time of treatment of the presentations that were considered viable, a cost estimate was made for the production of the presentations. With all the information obtained by the present study, it was estimated that the annual investment of US$ 1,951,966.91 (one million, nine hundred and fifty-one thousand, nine hundred and sixty-six US dollars and ninety-one cents) is necessary

Table 3. Annual cost estimate of the magistral production of selected drugs (Brazil, 2021).

| Presentations | Main Indication of Use | Average Price (USD) | Estimated Users | Average Treatment Time | Annual Cost Estimate (USD) |
|---|---|---|---|---|---|
| Chloramphenicol 25 mg/mL Oral solution | Alternative treatment of serious infections, by susceptible bacteria; Typhoid fever; Meningeal plague; Brazilian spotted fever and other rickettsiosis; Secondary infection by bothropic accident. | 7.56 | 29 | 1 month | 1,173.63 |
| Chloroquine phosphate 50 mg/5 mL Oral solution | Treatment of acute malaria caused by P. malaria, P. vivax and P. ovale (for infections caused by P. vivax and P. ovale, there is subsquent use of primaquine to eliminate intrahepatic forms); malaria prophylaxis; rheumatic disorders. | 7.23 | 10 | 12 months | 4,641.60 |
| Dapsone 25 mg/ml Oral solution | Leprosy treatment; Primary and secondary prophylaxis of Pneumocystis jirovecii pneumonia in co-infection with HIV (alternative regimen); Primary and secondary prophylaxis of Toxoplasma gondii, in co-infection with HIV (alternative regimen); Treatment of dermatitis herpetiformis. | 7.81 | 604 | 6 months | 151,483.20 |
| Doxycycline 25 mg/5 mL Oral solution | Bacterial infections; quinine or artesunate treatment supplement for multidrug treatment; P. falciparum malaria; short-term prophylaxis of multidrug-resistant P. falciparum Malaria. | 8.45 | 10 | 1 month | 452.00 |
| Ethambutol 25 mg/mL Oral solution | Treatment of sensitive tuberculosis, in association with other antimicrobials. Treatment of drug-resistant tuberculosis in association with other antimicrobials. Treatment of tuberculosis in cases of intolerance, allergy or toxicity to the basic regimen, in association with other antimicrobials; Treatment of tuberculosis-HIV co-infection replacing the rifampicin regimen, when associated with a protease inhibitor or integrase inhibitor; Treatment and secondary prevention of Mycobacterium avium complex infection in HIV positive patients. | 7.04 | 59 | 12 months | 26,656.20 |
| Leucovorin 5 mg/mL, Oral solution | Prevention of toxicity or treatment of pyrimethamine overdose; Methotrexate high-dose rescue or overdose treatment. | 8.79 | 2.700 | 12 months | 1,524,420.00 |
| Hydroxyurea 100 mg/mL Oral solution | Sickle Cell disease; Chronic myeloid leukemia | 14.07 | 5.000 | 12 months | 4,518,000.00 |
| Isoniazid + Rifampin 75 mg + 150 mg Capsule | Treatment of sensitive tuberculosis; Tuberculosis in cases of intolerance, allergy or toxicity to the basic regimen, in association with other antimicrobials. | 4.12 | 12.274 | 6 months | 1,624,586.64 |
| Pyrimethamine 1mg/ml Oral solution | Treatment of toxoplasmosis, in combination with sulfadiazine or clindamycin and leucovorin. | 7.38 | 2.700 | 12 months | 1,279,152.00 |
| Levofloxacin 250 mg/5ml Oral solution | Treatment of drug-resistant tuberculosis in association with other antimicrobials. Treatment of tuberculosis in cases of intolerance, allergy or toxicity to the basic regimen, in association with other antimicrobials. | 7.56 | 29 | 1 month | 1,173.63 |
| Sulfadiazine 100mg/5ml Oral solution | Toxoplasmosis; Prevention of recurrence of rheumatic fever; Uncomplicated acute urinary tract infections. | 7.57 | 2.700 | 12 months | 1,312,848.00 |
| **TOTAL** | | - | **26.115** | - | **1,951,966.91** |

Source: Elaborated by the authors, 2021.

for the provision of adequate treatment and, consequently, a better quality of life for approximately 26,115 people, mainly children, that is, an average of approximately US$ 74.75/year for each user.

It is noteworthy that 100% (n = 11) of the presentations for which the study obtained all the information are indicated for diseases whose responsibility for acquiring the respective treatments lies with the Strategic Component of Pharmaceutical Assistance, such as Leprosy, Tuberculosis, Toxoplasmosis and Sickle Cell Disease. Thus, it is observed that the total amount needed for the annual production of the listed presentations corresponds to only 3% of the budget approved for 2021 for the availability of medicines and health supplies for this

Component, which was of US$ 65,361.91 (sixty-five thousand, three hundred and sixty-one US dollars and ninety-one cents) (Table 3) [34, 35].

## Discussion

The unavailability of pharmaceutical presentations suitable for the pediatric public evidenced in this study may suggest that this situation encourages the use of products in manners that differ from those of their license (the so called off-label use) and/or the adaptation of pharmaceutical forms to obtain a more adequate dosage and/or presentation for administration [8, 11]. It should be considered, however, that this type of adaptation, especially when done in one's home, can lead to a loss of stability, with consequences for safety and effectiveness.

This relationship is corroborated by evidence of the use of "off-label" medications in pediatrics as a global reality.It is estimated that two thirds of the pediatric population use medications that do not have specific indications for their age group [8]. In Brazil, some studies point to a prevalence of around 51% in the use of off-labeldrugs in premature neonates, with systemic antiinfectives being the most used pharmaceutical specialty [8].

A study carried out in neonatal intensive care units in southern Italy showed a prevalence of 46.5% of the use of off-label drugs, with medicines containing furosemide, which are not authorized in the country for newborns, being the most prescribed [6]. A similar result was found at the Children's University Hospital in Bratislava and at the Pathological Newborns Unit of the Nitra University Hospital, in the Slovak Republic, where the prevalence was of 43% in the use ofoff-label medications. A high rate for off-label prescriptions was also reported in India, where the prevalence was of 51% [7, 8].

In the European Union (EU), in 2006, Pediatric Regulation No. 1901/2006 was approved, which aimed to reduce the need for the use of off-label drugs in pediatric pharmacotherapy, obliging companies to report research results on drug studies, to study drugs for the pediatric population and to develop age-appropriate formulations for all new drugs submitted for EU marketing authorization, including patent line extensions for protected drugs. However, a study carried out in Finland demonstrated a significant increase in the proportion of patients who received prescriptions for off-label use in 2011, in comparison to a similar study carried out in 2001, that is, before the approval of the regulation aimed at reducing this practice [5].

Children and adults have different therapeutic demands and these differences are not only due to pharmacokinetic and pharmacodynamic reasons, but also due to the need to adjust the dosage and obtain presentations with physical and organoleptic characteristics that allow acceptance by this public [10, 36].

Changes in drug absorption and metabolism that occur in different age groups in childhood should also be taken into account when choosing the dosage, the pharmaceutical presentation and the route of administration of the drug. Usually, in pediatric use, due to dysphagia, preference is given to liquid dosage forms, especially for neonates and babies [28], a fact that corroborates the present study, in which 82% of the drugs evaluated as priority for patient care referred to offering oral liquids.

It is noteworthy that, in the case of oral formulations used in pediatrics, the composition of the formulations must be evaluated, as some substances have restrictions on their use, especially in neonates. Benzoic acid, for example, should only be used from three years of age onwards due to incomplete metabolism in children up to this age group, which can cause toxicity to the nervous system. Other potentially restricted substances in pediatrics include benzyl alcohol due to neurotoxicity, metabolic acidosis and Gasping syndrome; polysorbate 20 and 80 can lead to liver and kidney failure; polyethylene glycol has the potential to cause metabolic acidosis, and azo compounds can cause urticaria, bronchoconstriction and angioedema [28].

These factors reinforce the importance of magistral production, guided by standard formulations previously defined in the literature, which establish parameters such as security and stability.

The absence of standard formulations was, in this study, mainly related to diseasespoverty-related. This fact confirms the importance of a policy to encourage the production of medicines that prioritize these diseases. Most of the presentations whose standard formula is not established in the consulted literature are intended for diseases whose treatment is performed exclusively in SUS, as is the case of tuberculosis and leprosy. The incentive for studies on the development of new formulations must be an initiative of public bodies, considering that the guarantee of adequate treatments is in the public and collective interest, with an impact on the improvement of the quality of life of the population neglected by the pharmaceutical industry.

The vast majority of pharmaceutical presentations unavailable in the Brazilian market could be produced and offered locally, in view of the availability of the active ingredient in the country, or obtained by adapting presentations different from those considered ideal but which constitute the same drug, such as case of Hydroxyurea used in the treatment of Sickle Cell Disease, which affects more than 25 thousand people annually, mainly the black population and people in situations of social vulnerability [1].

Due to the unavailability of Hydroxyurea in the oral solution presentation, there is currently a need to adapt solid drugs to oral liquids without, however, there being a regulation of this type of production bySUS. This change is performed most of the time by caregivers, generating risk because it is a cytotoxic drug, in addition to allowing errors in the administered dose due to the difficulty in understanding the dilution methodology [37].

At home manipulaiton is in disagreement with the regulation for good manipulation practices established by Anvisa Resolution RDC nº 67/2007, which provides for Good Manipulation Practices for Magistral and Official Preparations for Human Use in pharmacies. According to this regulation, the adaptation of pharmaceutical presentations, in order to adjust them to the prescription, can be carried out by pharmacies; however, the procedure must be justified in the manipulation order, weighing form or other internal quality document, preferably with technical justification or based on scientific literature, ensuring the quality of the drug in the new pharmaceutical form [38]. For this, it is essential to be careful at all stages of the process, such as in the acquisition of the pharmaceutical specialty to be transformed, which can be brought to the pharmacy by the user, being the pharmacist responsible for evaluating the conditions of this drug as well as its origin [38].

Despite the lack of a public policy to encourage and to finance the production of manipulated medicines in SUS, the practice is already used in some health units through specific initiatives of states and municipalities, which acquire the medicines through public bidding or pharmaceutical production, the latter being a more common practice in hospital units [13]. Among the manipulation experiences in the public network, the one carried out in the city of Ribeirão Preto stands out, where the Pharmaceutical Manipulation Laboratory produces 46 presentations for SUS patients [39]. For the Tuberculosis Program, 3% sodium chloride was produced for inhalation, used as a stimulus for pulmonary secretions before collecting material for examination, for the Leprosy program, an ethyl ether:ethane mixture was produced, used to test the sensitivity of the skin on regions affected by the disease [39]. The vast majority of the presentations produced, however, are for topical use, such as 2% acetic acid solution and sunscreen [39].

In some hospitals in the cities of Fortaleza, Porto Alegre, Niterói and Belo Horizonte, medications are also manipulated [40]. In the case of the Hospital of Belo Horizonte, the production of the drugs Captopril and Lorazepam stands out, which are not available in low doses or in liquid oral pharmaceutical forms [41]. At the Antônio Pedro University Hospital (Huap), in

the state of Rio de Janeiro, in a period of approximately two years, 657 formulations were produced through manipulation, with approximately 70% (457) destined for the Neonatal-ICU sector [40]. In these units, pharmaceutical manipulation corresponds to the preparation of unlicensed drugs to meet the specific needs of patients who do not have a drug in adequate presentation available on the market [42]. These experiences confirm that manipulation consists of a strategy to promote access to adequate pharmaceutical presentations, and that there is an urgent need of proposing a public policy to regulate this practice in SUS.

In Colombia, the National Institute of Cancerology (INC), interested in the possibility of producing magistral medicines and their potential application to increase access, in low and middle-income countries, to expensive biopharmaceuticals, started the implementation of masterful drug production. In 2017, the net income related to the production of radiopharmaceuticals reached about 1.2 million dollars and returned to INC profits, more than 250 thousand dollars, the equivalent to 20% of the indirect costs of the operation. Such experience corroborates the possibility of obtaining self paying technologies via manipulation with cost reduction and, consequently, access expansion [43].

In countries such as Portugal, Spain and Canada, the number of magistral pharmacies has been growing, mainly due to the advantages of personalized therapy, making manipulated drugs an important therapeutic tool, which enables the adequacy of doses, the association of drugs and the choice of the most suitable pharmaceutical presentation for the patient [11, 41]. The production of compounded medicines in community pharmacies is an important factor in public health, and in some regions it can reach 90% of community pharmacies that carry out this type of preparation [44].

In Brazil, according to data from the National Association of Magistral Pharmacists (Anfarmag), there are currently approximately 8,000 compounding pharmacies in the private network [45]. In regards to Higher Education Institutions (HEIs), of the 34 HEIs that are, or were, effective members of the National Forum of University Pharmacies—FNFU, 70.6% have a University Pharmacy, of which 70.8% perform drug manipulation (personal information)*. Studies show that the consumer confidence index in manipulated medicines is high, reaching 93% in some locations [46, 47]. Thus, considering that among the main factors for therapeutic success are the patient's access and reliability to the treatment and, consequently, the adherence to this treatment, these findings reinforce the hypothesis that magistral drug production can be an excellent alternative in the short and medium term to remedy the gaps in therapeutic care in SUS.

The main difficulty encountered in the study was the survey of the estimate of national demand for use in estimating the cost related to the development of manipulated drugs to mitigate the identified therapeutic gaps. Data on prevalence, incidence or number of cases related to indications for the use of presentations listed as priorities could not be accessed even through a request for information from the Ministry of Health through the Access to Information Law. This is because epidemiological surveillance is not established at the national level for some eliminated or rare diseases. Despite this limitation, the study observed that the investment amount would correspond to only 3% of the budget for the purchase of medicines financed exclusively by the Ministry of Health via the Strategic Component of Pharmaceutical Assistance (CESAF), having as parameter the approved budget for 2021.

However, it is noteworthy that the importance of offering manipulated drugs is not restricted to economic issues, as the strategy can provide greater versatility, providing therapy customization, which would meet what is proposed in this study in the case of pediatric presentations.

In this perspective, this study demonstrates that the inclusion of manipulated drugs in the National Medicines Policy is a plausible strategy for the correction of therapeutic and access gaps existing in SUS, presenting technical-operational and economic feasibility, and acting

where the pharmaceutical industry does not have a commercial interest, or does not have a scalability for industrial production, guaranteeing universal access to medicine and defending the right to health and to life.

Nonetheless, despite the feasibility of magistral drug production being able to fill most of the gaps in access to essential medicines identified as priorities, more studies are needed to deepen this topic, such as defining strategies for making available currently non-marketed active ingredients in the country, since 10% of the required active ingredients are not available in Brazil (Fig 1), developing formulations that are currently not available in the literature, as 14% of prioritized presentations do not have formulations described in the literature (Fig 2), expanding the demand analysis, pertinent and that have impact on other items not prioritized at the moment, since the study evaluated 30% of the 235 presentations identified as capable of magistral production. Furthermore, other assessments not addressed in this study should be carried out, such as the evalution of the structures of the units that would carry out the manipulation, a survey on the capillarity of the network and distribution logistics; and mainly, the elaboration of guidelines for financing and incorporating magistral drug production into SUS, as well as domains already addressed by other research, such as market sizing, market entry, financing, human and technical resources, including quality assurance and control [48].

## Conclusion

In the present study, it was possible to observe thatdespite the various initiatives carried out in Brazil and in the world, some therapeutic gaps persist, especially for diseases considered neglected, which makes the available treatment options very limited.

This reality also affects the pediatric population, challenging health professionals to offer an adequate pharmacotherapeutic to these patients and encouraging dose adjustment through non-recommended practices due to the insufficiency of safety studies and guarantee of efficacy.

Altogether, 56 priority drugs products that could be produced by manipulation were identified in this study, which would improve the quality of life of thousands of people who currently do not have access to drugs with adequate presentation for their condition.

Regarding the investment estimate, the annual value for the production of medicines was approximately US$ 74.75 per capita, representing, in its entirety, only 3% of the budget approved for 2021 for the Strategic Component of Pharmaceutical Assistance. This finding demonstrates that there is feasibility of incorporation of magistral production into SUS. In this context, it is urgent to consider magistral production as a plausible strategy for the correction of therapeutic and access gaps that exist in SUS, and, for this strategy to be included in the capacity of public health policies. It is necessary to implement actions that enable the development of formulations that are not yet available and identified as priorities for this study and to deepen what was initiated in this study.

After the implementation of magistral drug production as a public policy, the list of medicines must be constantly revised, in order to identify new gaps in national production, as well as to exclude drugs that have begun being industrially produced. These measures are important so that the policy of manipulated medicines is maintained, acting where the pharmaceutical industry does not have a commercial interest or where industrial production is not scalable, guaranteeing universal access to essential medicines.

## Author Contributions

**Conceptualization:** Gabriel Gonçalves Okamoto, Kathiely Martins dos Santos, Luma de Lira Nogueira, Guilherme Martins Gelfuso, Rafael Santos Santana.

**Data curation:** Gabriel Gonçalves Okamoto, Kathiely Martins dos Santos, Luma de Lira Nogueira, Guilherme Martins Gelfuso, Rafael Santos Santana.

**Formal analysis:** Gabriel Gonçalves Okamoto, Kathiely Martins dos Santos, Luma de Lira Nogueira, Guilherme Martins Gelfuso, Rafael Santos Santana.

**Supervision:** Rafael Santos Santana.

**Validation:** Guilherme Martins Gelfuso.

**Writing – original draft:** Gabriel Gonçalves Okamoto, Kathiely Martins dos Santos, Luma de Lira Nogueira.

**Writing – review & editing:** Guilherme Martins Gelfuso, Rafael Santos Santana.

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
