## [Decision Letter · Decision Letter 0]

1 Sep 2022

PONE-D-22-14777Medicine manipulation: an alternative to mitigate therapeutic gaps in the Brazilian Unified Health System?PLOS ONE

Dear Dr. Santana,

Thank you for submitting your manuscript to PLOS ONE. After careful consideration, we feel that it has merit but does not fully meet PLOS ONE’s publication criteria as it currently stands. Therefore, we invite you to submit a revised version of the manuscript that addresses the points raised during the review process.

We look forward to receiving your revised manuscript.

Kind regards,

Paola Brusa

Academic Editor

PLOS ONE

Journal Requirements:

Additional Editor Comments (if provided):

I thank the authors for reading their paper: masgistral preparations are still very necessary in many countries.

The paper need minor revisions before to be publish.

Reviewers' comments:

Reviewer's Responses to Questions

**Comments to the Author**

1. Is the manuscript technically sound, and do the data support the conclusions?

Reviewer #1: Yes

Reviewer #2: Yes

2. Has the statistical analysis been performed appropriately and rigorously? 

Reviewer #1: Yes

Reviewer #2: N/A

3. Have the authors made all data underlying the findings in their manuscript fully available?

Reviewer #1: Yes

Reviewer #2: Yes

4. Is the manuscript presented in an intelligible fashion and written in standard English?

Reviewer #1: Yes

Reviewer #2: Yes

5. Review Comments to the Author

Reviewer #1: The manuscript is acceptable in the current form. The authors have used multiple methods to provide an detailed methodology, analysis, and results on medical manipulation for mitigating gaps in therapeutic provision in the Brazilian Health System.

Reviewer #2: This is a novel, important breakthrough paper. The objective is honorable in securing treatments needed for many diseases of poverty and in children.

The abstract does not define the SUS acronym and it would be better referred to in general as the Brazil health system in the abstract. When the abstract describes “national production” it should specify Brazil as the nation.

The abstract is missing some of the most exciting motivations and results of the paper. You could add for example "The study prioritized candidate drugs for cost-effective local manufacturing in a Brazil laboratory to fill unmet needs in the population... We found the absence of a standard formula for production was most often in indications for poverty-related diseases."

The terminology of “magistral medicines”, “magistral production”, and “manipulation” need to be defined early in the introduction. Do the authors refer to production at a central facility or does this also include manipulation at the pharmacy level? The abstract should also provide a short definition before using these terms.

This sentence from the results/discussion should be moved to the beginning of the introduction section. “In Brazil, the National Health Surveillance Agency (ANVISA) is the agency responsible for authorizing drug registration in the national territory. However, so far there is no specific regulation for medications intended for the pediatric population, nor are there any evidence-based guidelines on the transformation of pharmaceutical forms for the administration of adequate doses.”

Results lines 226-228 should be closer to the beginning of the introduction because they are very motivating for readers. The use for kids, tube users, swallowing difficulties for “therapeutic orphans” is a good explanation right after you define the concepts of magistral production.

Add citation of other papers that provide examples of magistral production in middle-income countries, such as this one from Colombia:

Vaca González, C.P., Arteaga, L. & Delgado López, N.E. Magistral drug production in Colombia and other middle-income countries. Nat Biotechnol 37, 216–217 (2019). https://doi.org/10.1038/s41587-019-0044-z. https://www.nature.com/articles/s41587-019-0044-z

It is confusing when the authors refer to availability “in Brazil and in SUS” and could be more clear to readers rephrased as “available in the country and covered by the health system.” This applies in at least lines 72, 80, 98, 101-105.

The section on categorization of preparations by low vs high complexity and by active ingredients commercially available vs not available in Brazil can reference this prior work establishing those concepts as dimensions to evaluate to achieve the objectives of this paper:

Zimmermann M, Adamson B, Lam-Hine T, Rennie T, Stergachis A. Assessment tool for establishing local pharmaceutical manufacturing in low- and middle-income countries. Int J Pharm Pract. 2018 Aug;26(4):364-368. doi: 10.1111/ijpp.12455. Epub 2018 May 6. PMID: 29732641.

Consider working with the editor or layout designer to create a “call-out” or “highlighted box” titled “Therapeutic Gaps Considered” with three bullet points from manuscript lines 101-105 in the methods section. This is key to understanding the paper and readers may want to refer to it when interpreting each section of the paper.

Can you separate the results section from the discussion section? The results should contain only new information that was learned in the study and the discussion can include the interpretation of the results, examples from other countries, and discuss the implications for policy and manufacturing in Brazil.

Results line 203 would be clearer rephrased “Among the essential medicines needed to fill treatment gaps in Brazil, we identified 56 drug products that could be locally manufactured.”

Consider a new title for Table 1 as “Summary of medicine types with potential for manufacturing in Brazil, according to ATC type.” Define ATC acronym in the footnote of the table. Be more specific in the table column headers (quantity of what? percentage of what?)

How were the 56 drugs in Table 2 selected from the total number that the authors say were found to be eligible 56% of 253? How were the drugs in Table 3 selected from the longer list in Table 2?

It would assist readers if the results section about costs could include a conversion of the total amount into USD $ for easier interpretation globally.

The manuscript methods description of costing used in the study was clear but it was not clear in Table 3 how to interpret the total costs for each drug. Is this the estimated value or budget impact if locally manufactured?

The phrase “susceptible to manipulation” makes sense to pharmacists but I fear will be misunderstood by the general audience reading the manuscript. Is there an alternative way to describe this attribute using other words that will be more effective for clear communication? For example, “drug products that are plausible candidates for local manufacturing given low complexity of formulations using active ingredients commercially available in Brazil.”

The text inside Figures need to be translated into English.

Please copyedit the decimal and comma notation for numbers within tables so they align with PLOS ONE style guidelines.

6. PLOS authors have the option to publish the peer review history of their article (what does this mean?). If published, this will include your full peer review and any attached files.

Reviewer #1: **Yes: **Denny John

Reviewer #2: No

---

## [Author Response · Author response to Decision Letter 0]

10 Oct 2022

Reviewer #1:

1) The manuscript is acceptable in the current form. The authors have used multiple methods to provide an detailed methodology, analysis, and results on medical manipulation for mitigating gaps in therapeutic provision in the Brazilian Health System.

R: We thank the reviewer for positively evaluating our manuscript.

Reviewer #2:

1) This is a novel, important breakthrough paper. The objective is honorable in securing treatments needed for many diseases of poverty and in children.

R: We thank the reviewer for the positive evaluation of our manuscript. Your suggestions were considered in the elaboration of the revised manuscript, which we believe have contributed to the improvement of its quality.

2) The abstract does not define the SUS acronym and it would be better referred to in general as the Brazil health system in the abstract. When the abstract describes “national production” it should specify Brazil as the nation.

R: In response to this recommendation, the definition of the SUS acronym is now included in line 20. We also included the explanation that the study was carried out considering the Brazilian sanitary context (line 22).

3) The abstract is missing some of the most exciting motivations and results of the paper. You could add for example "The study prioritized candidate drugs for cost-effective local manufacturing in a Brazil laboratory to fill unmet needs in the population... We found the absence of a standard formula for production was most often in indications for poverty-related diseases."

R: The abstract was properly reviewed and the main motivations for the study were included in lines 20-23 and 27-29. In addition, we added information regarding candidate drugs for local manufacture that were prioritized in the evaluation of production costs, considering the Brazilian health context. We reiterate the fact that the absence of studies to develop a standard formula was, in this study, related to drugs that had therapeutic indications for poverty-related diseases.

4) The terminology of “magistral medicines”, “magistral production”, and “manipulation” need to be defined early in the introduction. Do the authors refer to production at a central facility or does this also include manipulation at the pharmacy level? The abstract should also provide a short definition before using these terms.

R: We thank the reviewer for the suggestion. We have changed the term to "magistral drug production" throughout the manuscript.

5) This sentence from the results/discussion should be moved to the beginning of the introduction section. “In Brazil, the National Health Surveillance Agency (ANVISA) is the agency responsible for authorizing drug registration in the national territory. However, so far there is no specific regulation for medications intended for the pediatric population, nor are there any evidence-based guidelines on the transformation of pharmaceutical forms for the administration of adequate doses.”

R: We thank you for your contribution. The sentence is now included in the introduction section (lines 49-54) in the revised version of the manuscript.

6) Results lines 226-228 should be closer to the beginning of the introduction because they are very motivating for readers. The use for kids, tube users, swallowing difficulties for “therapeutic orphans” is a good explanation right after you define the concepts of magistral production.

R: The introduction was properly altered. Please, check lines 41-45.

7) Add citation of other papers that provide examples of magistral production in middle-income countries, such as this one from Colombia:

Vaca González, C.P., Arteaga, L. & Delgado López, N.E. Magistral drug production in Colombia and other middle-income countries. Nat Biotechnol 37, 216–217 (2019). https://doi.org/10.1038/s41587-019-0044-z. https://www.nature.com/articles/s41587-019-0044-z

R: The experience carried out in Colombia was included in this revised version of the manuscript, between lines 423 and 430.

8) It is confusing when the authors refer to availability “in Brazil and in SUS” and could be more clear to readers rephrased as “available in the country and covered by the health system.” This applies in at least lines 72, 80, 98, 101-105.

R: We appreciate this contribution. The terminology “medicines covered by SUS”. Such adjustments can be found in lines 79,106, 108, 113-116.

9) The section on categorization of preparations by low vs high complexity and by active ingredients commercially available vs not available in Brazil can reference this prior work establishing those concepts as dimensions to evaluate to achieve the objectives of this paper:

Zimmermann M, Adamson B, Lam-Hine T, Rennie T, Stergachis A. Assessment tool for establishing local pharmaceutical manufacturing in low- and middle-income countries. Int J Pharm Pract. 2018 Aug;26(4):364-368. doi: 10.1111/ijpp.12455. Epub 2018 May 6. PMID: 29732641.

R: We appreciate the contribution. The experience carried out in the Zimmermann et al. study was referenced in lines 480-482.

10) Consider working with the editor or layout designer to create a “call-out” or “highlighted box” titled “Therapeutic Gaps Considered” with three bullet points from manuscript lines 101-105 in the methods section. This is key to understanding the paper and readers may want to refer to it when interpreting each section of the paper.

R: In response to the suggestion, we decided instead of including a figure in the manuscript, to make a subdivision in the methods to improve the understanding of the readers (lines 90 - 124).

11) Can you separate the results section from the discussion section? The results should contain only new information that was learned in the study and the discussion can include the interpretation of the results, examples from other countries, and discuss the implications for policy and manufacturing in Brazil.

R: Given the suggestion, we divided the sections: the results are now available between lines 218 and 318 and the between lines 319 and 482.

12) Results line 203 would be clearer rephrased “Among the essential medicines needed to fill treatment gaps in Brazil, we identified 56 drug products that could be locally manufactured.”

R: Aiming at improving the quality of the manuscript, we adjusted the text between lines 293 and 296, making it clearer that the medicines would be produced locally, and in which structures these medicines could be produced.

13) Consider a new title for Table 1 as “Summary of medicine types with potential for manufacturing in Brazil, according to ATC type.” Define ATC acronym in the footnote of the table. Be more specific in the table column headers (quantity of what? percentage of what?)

R: We appreciate the suggestion. A new title was included in Table 1, available in lines 234-235. In addition, the table header was also altered, and the acronym ATC was defined in a footnote following the table, as suggested.

14) How were the 56 drugs in Table 2 selected from the total number that the authors say were found to be eligible 56% of 253? How were the drugs in Table 3 selected from the longer list in Table 2?

R: As described in the methods section, after the identification of drugs that could be produced via manipulation, SUS demands were prioritized, where of the 235 drugs identified as feasible to be produced via manipulation, 30% were identified as a priority, i.e., 70 drug products.

Based on the priority drugs (n=70), a theoretical assessment of the feasibility of producing these drugs via manipulation was carried out, considering that drugs that had both the standard formula described in the literature and the availability of the active principle or of a finished product available in the Brazilian market were considered viable for production by manipulation. Thus, table 2 presents the list of 56 drugs that were considered feasible, making them suitable for the cost estimate evaluation.

In relation to Table 3, the prioritized drugs, which presented theoretical and production feasibility (n=56), had their cost survey for local production estimated. To estimate demand, data on prevalence, incidence or when unavailable, number of cases in 2019, of diseases and conditions for which the prioritized presentations have a main indication were considered, according to the age group of the target audience. However, as described in the study, one of the main limitations found was the survey of the demand estimate, which made it impossible to present the demand estimate for some drugs.

Table 3 presents the annual cost estimate of local production, through the handling of medicines whose study carried out the survey of the cost estimate, when it identified the demand estimate.

15) It would assist readers if the results section about costs could include a conversion of the total amount into USD $ for easier interpretation globally.

R: We thank the reviewer for the suggestion. The conversion of all values in Brazilian Real to US Dollars was carried out.

16) The manuscript methods description of costing used in the study was clear but it was not clear in Table 3 how to interpret the total costs for each drug. Is this the estimated value or budget impact IF locally manufactured?

R: New wording was included between lines 293 and 296 to improve the interpretation of the data from the Table 3.

17) The phrase “susceptible to manipulation” makes sense to pharmacists but I fear will be misunderstood by the general audience reading the manuscript. Is there an alternative way to describe this attribute using other words that will be more effective for clear communication? For example, “drug products that are plausible candidates for local manufacturing given low complexity of formulations using active ingredients commercially available in Brazil.”

R: Firstly, we thank you for your contribution. Aiming at a better understanding by the general public, the adaptation suggested was carried out in lines 237-238, changing the terminology to ¨drugs identified as plausible candidates for local manufacture by manipulation¨.

18) The text inside Figures need to be translated into English.

R: Firstly, the text into the Figures were translated. We are sorry for the mistake.

19) Please copyedit the decimal and comma notation for numbers within tables so they align with PLOS ONE style guidelines.

R: We performed the suggested change in Table 3.

---

## [Editor Report · Decision Letter 1]

14 Oct 2022

Medicine manipulation: an alternative to mitigate therapeutic gaps in the Brazilian Unified Health System?

PONE-D-22-14777R1

Dear Dr. Santana,

We’re pleased to inform you that your manuscript has been judged scientifically suitable for publication and will be formally accepted for publication once it meets all outstanding technical requirements.

Kind regards,

Paola Brusa

Academic Editor

PLOS ONE

---

## [Editor Report · Acceptance letter]

18 Oct 2022

PONE-D-22-14777R1 

Medicine manipulation: an alternative to mitigate therapeutic gaps in the Brazilian Unified Health System?Medicine manipulation: an alternative to mitigate therapeutic gaps in the Brazilian Unified Health System? 

Dear Dr. Santana:

I'm pleased to inform you that your manuscript has been deemed suitable for publication in PLOS ONE. Congratulations! Your manuscript is now with our production department. 

Kind regards, 

on behalf of

Professor Paola Brusa 

Academic Editor

PLOS ONE